# Lung-protective properties of expiratory flow-initiated pressure-controlled inverse ratio ventilation: A randomised controlled trial

Go Hirabayashi *, Minami Saito, Sachiko Terayama, Yuki Akihisa, Koichi Maruyama, Tomio Andoh

Department of Anaesthesiology, Mizonokuchi Hospital Teikyo University School of Medicine, Kanagawa, Japan

* goh@med.teikyo-u.ac.jp

**Data Availability Statement:** All relevant data are within the paper and its Supporting Information files.

## Abstract

### Background

Expiratory flow-initiated pressure-controlled inverse ratio ventilation (EF-initiated PC-IRV) reduces physiological dead space. We hypothesised that EF-initiated PC-IRV would be lung protective compared with volume-controlled ventilation (VCV).

### Methods

Twenty-eight men undergoing robot-assisted laparoscopic radical prostatectomy were enrolled in this randomised controlled trial. The EF-initiated PC-IRV group (n = 14) used pressure-controlled ventilation with the volume guaranteed mode. The inspiratory to expiratory (I:E) ratio was individually adjusted by observing the expiratory flow-time wave. The VCV group (n = 14) used the volume control mode with a 1:2 I:E ratio. The Mann–Whitney U test was used to compare differences in the serum cytokine levels.

### Results

There were no significant differences in serum IL-6 between the EF-initiated PC-IRV (median 34 pg ml$^{-1}$ (IQR 20.5 to 63.5)) and VCV (31 pg ml$^{-1}$ (24.5 to 59)) groups ($P = 0.84$). The physiological dead space rate (physiological dead space/expired tidal volume) was significantly reduced in the EF-initiated PC-IRV group as compared with that in the VCV group ($0.31 \pm 0.06$ vs $0.4 \pm 0.07$; $P<0.001$). The physiological dead space rate was negatively correlated with the forced vital capacity (% predicted) in the VCV group ($r = -0.85$, $P<0.001$), but not in the EF-initiated PC-IRV group ($r = 0.15$, $P = 0.62$). Two patients in the VCV group had permissive hypercapnia with low forced vital capacity (% predicted).

### Conclusions

There were no differences in the lung-protective properties between the two ventilatory strategies. However, EF-initiated PC-IRV reduced physiological dead space rate; thus, it may be useful for reducing the ventilatory volume that is necessary to maintain normocapnia

**Funding:** This study was financially supported by internal funding from the Department of Anaesthesiology, Mizonokuchi Hospital Teikyo University School of Medicine, Kanagawa, Japan. The funders had no role in study design, data collection and analysis, decision to publish, or preparation of the manuscript.

**Competing interests:** The authors have declared that no competing interests exist.

in patients with low forced vital capacity (% predicted) during robot-assisted laparoscopic radical prostatectomy.

## Introduction

Patients are placed in a steep Trendelenburg position with $CO_2$ pneumoperitoneum during robot-assisted laparoscopic radical prostatectomy. This procedure can decrease lung functional residual capacity and lung-thoracic compliance and increase ventilation-perfusion mismatch [1–3]. Pressure-controlled inverse inspiratory to expiratory ratio ventilation has been used in acute respiratory distress syndrome [4–8]. However, its clinical utility remains controversial [9–12]. Earlier studies used pressure-controlled inverse ratio ventilation (PC-IRV) with an inspiratory to expiratory (I:E) ratio of 2:1 to 4:1 without individual adjustment; this resulted in a very short expiratory phase and increased the risk of lung hyperinflation and circulatory depression. PC-IRV with an IE ratio that is individually adjusted by observing the expiratory flow-time wave can appropriately maintain moderate total positive end-expiratory pressure (total PEEP). We previously reported that expiratory flow (EF)-initiated PC-IRV reduced physiological dead space ($VD_{phys}$) without lung hyperinflation and circulatory depression in patients undergoing robot-assisted laparoscopic radical prostatectomy [13]. Additionally, we hypothesised that EF-initiated PC-IRV may have lung-protective properties because this method can lower mechanical stress on the lung tissue by reducing $VD_{phys}$ and the ventilatory volume required to maintain the partial pressure of $CO_2$ ($PaCO_2$).

This study primarily aimed to evaluate the lung-protective properties of EF-initiated PC-IRV and to compare the differences in the lung-protective properties between EF-initiated PC-IRV and volume-controlled ventilation (VCV) in patients undergoing robot-assisted laparoscopic radical prostatectomy. The secondary aim was to compare the ventilatory efficacy of these methods.

## Materials and methods

### Study design and ethics

This study was designed as a prospective, mono-centre, single-blinded, randomized controlled clinical trial. It was approved by the Ethical Committee of Teikyo University School of Medicine, Tokyo, Japan on 12 September 2017 (No. 17–063) and was registered at UMIN Clinical Trials Registry (UMIN000029552). The study was conducted in the Mizonokuchi Hospital Teikyo University School of Medicine, Kanagawa, Japan, between December 2017 and September 2018. The original Japanese study protocol approved by the Ethical Committee of Teikyo University School of Medicine as the S1 File, and English protocol as the S2 File. This study is reported in adherence to the CONSORT guidelines, and the CONSORT checklist is provided in the S1 Checklist.

### Subjects

Patients aged 18 to 85 years with American Society of Anesthesiologists (ASA) physical status I or II and who were scheduled for robot-assisted laparoscopic radical prostatectomy were considered eligible for participation in this study. The exclusion criteria were as follows: ASA physical status 3 to 5, a history of pneumothorax, and previous lung surgery. Written informed consent was obtained from all eligible patients. The participants were recruited between December 2018 and September 2019.

## Randomization and blinding

Researchers at the Teikyo Academic Research Centre randomised patients to the VCV or EF-initiated PC-IRV groups with a 1:1 allocation ratio using an envelope method after generating the allocation sequence. Only the patients remained blinded during the whole study procedure.

## Anaesthesia protocol

Routine patient monitoring included electrocardiography, pulse oximetry, oesophageal temperature measurement, non-invasive arterial blood pressure measurement, and anaesthetic gas $CO_2$ analysis. Moreover, the Vigileo with the Flo-Trac sensor (Edwards Lifesciences, Irvine, CA, USA) was used to monitor continuous radial arterial pressure, cardiac index (*CI*), and stroke volume variation (SVV). Mainstream $CO_2$ and flow sensors were attached to the proximal end of the tracheal tube to enable volumetric capnography (Senko Medical Instrument Co., Ltd., Tokyo, Japan). Anaesthesia was induced by the administration of 1–3 mg kg$^{-1}$ of intravenous propofol and 2–4 μg kg$^{-1}$ of fentanyl. Tracheal intubation was performed with an 8.0-mm cuffed tube following the administration of 0.6–0.9 mg kg$^{-1}$ of rocuronium. Anaesthesia was maintained with volatile anaesthetic gas that was composed of 3–4% desflurane and supplemented with continuous intravenous infusions of 0.2–0.3 μg kg$^{-1}$ min$^{-1}$ remifentanil and intermittent intravenous injections of 0.1–0.2 mg kg$^{-1}$ rocuronium and 1–2 μg kg$^{-1}$ fentanyl when needed.

The same anaesthesia ventilator (Avance Carestation, Datex-Ohmeda, GE Healthcare, Helsinki, Finland) was used for all of the patients. The initial ventilator settings included the volume control (VC) mode, the tidal volume ($V_T$) that was set at 8–10 ml kg$^{-1}$ of the ideal body weight (IBW) (i.e., 50 + 0.91 × [height in cm—152.4]), a respiratory rate of 12 breaths min$^{-1}$, a baseline airway pressure (BAP) (used as set PEEP) of 5 cmH$_2$O, 0.5 fraction of inspired oxygen ($F_iO_2$), and an I:E ratio of 1:2.

## Interventions and ventilatory strategies

Ventilator settings were switched to the EF-initiated PC-IRV or VCV strategy (Table 1) following the establishment of the 25–30˚ Trendelenburg position and $CO_2$ pneumoperitoneum at 12 mmHg. The EF-initiated PC-IRV strategy included the pressure-controlled ventilation-volume guaranteed (PCV-VG) mode. In this mode, the airway pressure is adjusted to achieve a

**Table 1. Ventilator strategy.**

|  | VCV strategy | EF-initiated PC-IRV strategy |
|---|---|---|
| Ventilator mode | volume control with pause ratio of 20% | pressure-controlled ventilation-volume guaranteed |
| Inspiratory:Expiratory (ratio) | 1:2 | 2:1, 1.5:1, or 1:1 were selected so that inspiration was initiated at the midpoint between the expiratory flow change point and the return point to the expected baseline |
| Baseline airway pressure (cmH$_2$O) | 5 | 0 |
| Respiratory rate | The initial respiratory rate was 12 beats min$^{-1}$, allowed for an increase in the respiratory rate to an upper limit of 18 beats min$^{-1}$ to achieve a PaCO$_2$ of less than 50 mmHg | |
| Target tidal volume | Adjusted to achieve a plateau pressure upper limit of 30 cmH$_2$O | |

VCV, volume-controlled ventilation; EF-initiated PC-IRV, expiratory flow-initiated pressure-controlled inverse ratio ventilation.

target tidal volume, and plateau pressures are allowed to reach an upper limit of 30 cmH$_2$O. BAP was set off. The initial respiratory rate was 12 beats min$^{-1}$, and the I:E ratio was individually adjusted by observing the expiratory flow-time wave [13]. I:E ratios of 2:1, 1.5:1, or 1:1 were selected so that inspiration was initiated at the midpoint between the expiratory flow change point and the return point to the expected baseline. The VCV strategy included the VC mode, and a pause ratio of 20% was used in order to measure plateau pressure. A target tidal volume was set with a plateau pressure upper limit of 30 cmH$_2$O. BAP was set to 5 cmH$_2$O. The I:E ratio was 1:2, and the initial respiratory rate was 12 beats min$^{-1}$. Both strategies allowed for an increase in the respiratory rate to an upper limit of 18 beats min$^{-1}$ to achieve a $P$aCO$_2$ of less than 50 mmHg that was estimated from end-tidal CO$_2$ (E$_T$CO$_2$) changes and differences between E$_T$CO$_2$ and $P$aCO$_2$ on arterial blood gas analysis. Hypercapnia ($>$ 50 mmHg) was permitted if the respiratory rate increased to 18 beats min$^{-1}$ with a plateau pressure of 30 cmH$_2$O.

Immediately after returning the patients to the supine position and relieving CO$_2$ pneumoperitoneum, patients in the EF-initiated PC-IRV or VCV groups were switched back to the initial VCV mode. Haemodynamics were maintained throughout the study with a mean arterial pressure (MAP) >70 mmHg, $CI$ >2 l min$^{-1}$ m$^{-2}$, and an SVV <15%. If MAP fell below 70 mmHg, intravenous ephedrine (4–8 mg) was administered. If the SVV exceeded 15%, an additional intravenous fluid challenge was provided with 10 ml kg$^{-1}$ of Ringer's acetate solution or hydroxyethyl starch. Pulse oximetry-monitored oxygen saturation was allowed to drop to a lower limit of 93%. When these parameters exceeded the predetermined limits, alveolar recruitment manoeuvres consisting of applying a continuous positive airway pressure of 30 cmH$_2$O for 30 seconds were conducted, and the ventilator setting was changed by increasing the respiratory rate and F$_i$O$_2$ and increasing or decreasing the set tidal volume.

## Outcomes

The primary outcome was the change in serum IL-6 levels, which was used as a surrogate marker for both surgical- and ventilator-induced lung injury. The proinflammatory cytokines, IL-8 and IL-1β, were also evaluated [14]. Each cytokine was measured at T$_{Baseline}$ (20 min after the initial setting) and T$_{End}$ (end of surgery). The secondary outcome included VD$_{phys}$, which was calculated as [VD$_{phys}$ = V$_{TE}$ × ($P$aCO$_2$—$P_E$CO$_2$) $P$aCO$_2$$^{-1}$]. Expired tidal volume (V$_{TE}$) and expired CO$_2$ partial pressure ($P_E$CO$_2$) were measured with volumetric capnography. Static compliance (C$_{stat}$) was modified as [C$_{stat}$ = V$_{TI}$/(P$_{plat}$—PEEP)] (V$_{TI}$, inspired tidal volume measured by volumetric capnography; P$_{plat}$, plateau airway pressure). Driving pressure was calculated as (Driving pressure = P$_{plat}$–PEEP). Each respiratory or haemodynamic parameter was measured at T$_{Baseline}$ and T$_{2h}$ (2 h after intervention). The incidences of permissive hypercapnia (PaCO$_2$ > 50 mmHg) and respiratory complications were also recorded.

## Statistical analysis

Based on previous study data [14], the calculated sample size was 14 subjects per group to detect differences in IL-6 concentration between the ventilatory settings with the given one-sided test with a power of 0.8 and a type I error rate of 0.05 based on an estimated difference of 1 of the parameter's estimated SD.

All parametric data were described using mean (SD) and the non-parametric data using median (first and third quartiles). Parametric data comparisons were performed using Student's t-test and non-parametric data analyses were conducted using the Mann-Whitney U test. The association between serum cytokines and other parameters was evaluated using Spearman's correlation coefficient, and the association between the other continuous variables

was evaluated using linear regression analysis. *P*-values <0.05 were considered statistically significant. Statistical analyses were performed using EZR provided by the Comprehensive R Archive Network.

## Results

A detailed study flow chart is presented in Fig 1. The enrolment of patients started on 6 December 2017. A total of 39 consecutive patients who were undergoing robot-assisted laparoscopic radical prostatectomy were screened; five patients did not meet the inclusion criteria, and six patients refused consent. Written informed consent was obtained from the 28 eligible patients who were randomly divided into the following two groups: 1) the VCV group (n = 14) and 2) the EF-initiated PC-IRV group (n = 14). Thus, 14 patients from each group were analysed.

There were no significant differences in patient and surgical characteristics between the two groups (Table 2).

The I:E ratio in the EF-initiated PC-IRV group was 1.8 ± 0.2 (10 patients at 2:1 ratio and 4 patients at 1.5:1 ratio), and the I:E ratio in the VCV group was 0.5 (14 patients at 1:2 ratio). Although there were significant differences in $VD_{phys}/V_{TE}$ and SVV between the EF-initiated PC-IRV group and the VCV group, there were no significant differences in any of the other respiratory or haemodynamic parameters between the two groups (Table 3).

Further, there were no significant differences in serum IL-6 and IL-8 levels between the VCV and EF-initiated PC-IRV groups (Table 4).

Measurements of IL-1β were discontinued after 10 patients because no increases were detected. There was a significant positive correlation between serum IL-6 at $T_{End}$ and the duration of surgery in both the EF-initiated PC-IRV group (r = 0.82, P<0.001) and the VCV group

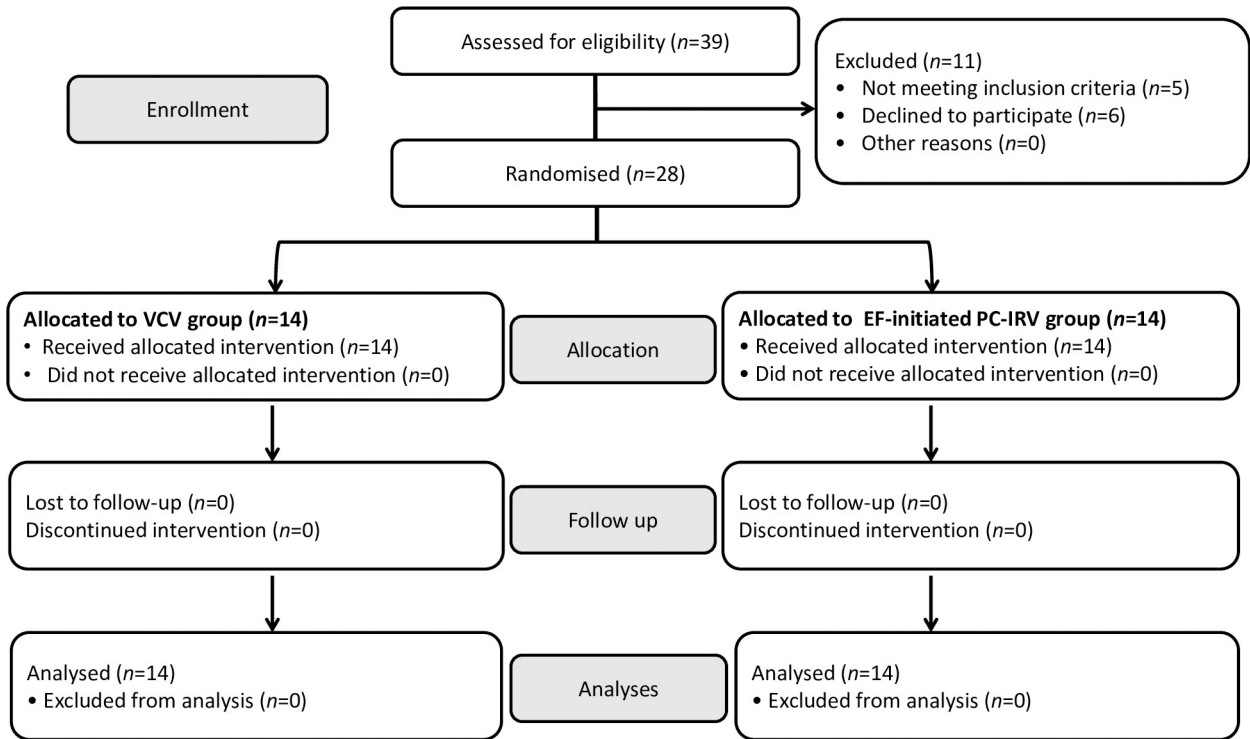

**Fig 1. CONSORT diagram of patient recruitment for the comparisons between volume-controlled ventilation (VCV) and expiratory flow-initiated pressure-controlled inverse ratio ventilation (EF-initiated PC-IRV).**

**Table 2. Patient and surgery characteristics.**

| | VCV group (n = 14) | EF-initiated PC-IRV group (n = 14) | [a]P-value |
|---|---|---|---|
| **Patient characteristics (all men)** | | | |
| Age (years) | 67.8 ± 5.9 | 69.3 ± 4.4 | 0.49 |
| Height (cm) | 168 ± 4.4 | 166 ± 5.4 | 0.33 |
| Weight (kg) | 70.9 ± 10.6 | 71.2 ± 11.5 | 0.8 |
| Body mass index (kg m$^{-2}$) | 25.1 ± 3.3 | 25.6 ± 3.6 | 0.71 |
| FVC % predicted (%) | 106 ± 14 | 108 ± 13 | 0.82 |
| FEV$_1$% predicted (%) | 78.1 ± 8.1 | 80.5 ± 6.4 | 0.43 |
| ASA physical status; 1/2/3 (n) | 4/10/0 | 2/12/0 | 0.65 |
| **Surgery characteristics** | | | |
| Duration of surgery (min) | 228 ± 64 | 204 ± 36 | 0.24 |
| Duration of the Trendelenburg position and pneumoperitoneum (min) | 191 ± 55 | 175 ± 43 | 0.41 |
| Perioperative blood loss (ml) | 260 ± 195 | 180 ± 139 | 0.45 |

Values are presented as the mean ± SD or numbers.

[a]P-values are from Fisher's exact test (qualitative data) or Mann-Whitney U test (quantitative data), respectively. VCV, volume-controlled ventilation; EF-initiated PC-IRV, expiratory flow-initiated pressure-controlled inverse ratio ventilation; FVC, forced vital capacity; FEV$_1$, forced expiratory volume in 1 second.

**Table 3. Ventilator settings and respiratory and haemodynamic variables.**

| Variable | Time period | VCV group | EF-initiated PC-IRV group | | 95% CI of the difference | [a]P-value |
|---|---|---|---|---|---|---|
| | | VCV strategy | VCV strategy | PC-IRV strategy | | |
| Respiratory rate (beats min$^{-1}$) | T$_{Baseline}$ | 12 | 12 | | | |
| | T$_{2h}$ | 13.4 ± 1.7 | | 13.2 ± 1.5 | -1.6 to 1 | 0.62 |
| Expired tidal volume (ml) | T$_{Baseline}$ | 522 ± 34 | 527 ± 32 | | -21.7 to 32.3 | 0.69 |
| | T$_{2h}$ | 453 ± 62$^†$ | | 471 ± 66$^†$ | -35.1 to 70 | 0.5 |
| Plateau pressure (cmH$_2$O) | T$_{Baseline}$ | 13.8 ± 1.5 | 14.4 ± 1.9 | | -8.3 to 2 | 0.41 |
| | T$_{2h}$ | 24.3 ± 2.9$^†$ | | 23.4 ± 2.9$^†$ | -3.3 to 1.4 | 0.42 |
| Driving pressure (cmH$_2$O) | T$_{Baseline}$ | 8.9 ± 1.5 | 9.4 ± 2 | | -0.83 to 1.97 | 0.41 |
| | T$_{2h}$ | 19.4 ± 2.9$^†$ | | 18.9 ± 2.5$^†$ | -2.95 to 1.95 | 0.68 |
| PaCO$_2$ (mmHg) | T$_{Baseline}$ | 36.6 ± 4 | 36.6 ± 2.9 | | -3.1 to 2.6 | 0.84 |
| | T$_{2h}$ | 46.3 ± 5.5$^†$ | | 43.8 ± 4.6$^†$ | -1.3 to 1.6 | 0.22 |
| E$_T$CO$_2$ (mmHg) | T$_{Baseline}$ | 33.3 ± 2.5 | 32.4 ± 2.9 | | -3.1 to 1.2 | 0.39 |
| | T$_{2h}$ | 39.3 ± 3.3$^†$ | | 38.7 ± 3.6$^†$ | -3.3 to 2.2 | 0.67 |
| PaO$_2$/F$_I$O$_2$ (ratio) | T$_{Baseline}$ | 409 ± 86 | 381 ± 93 | | -99.5 to 44.7 | 0.44 |
| | T$_{2h}$ | 374 ± 97 | | 365 ± 84 | -82.8 to 63.7 | 0.79 |
| Static compliance (ml cmH$_2$O$^{-1}$) | T$_{Baseline}$ | 63.5 ± 9.8 | 61 ± 11.3 | | -11 to 6.1 | 0.56 |
| | T$_{2h}$ | 25.2 ± 6.6$^†$ | | 26.2 ± 7.9$^†$ | -4.8 to 6.9 | 0.73 |
| VD$_{phys}$/V$_{TE}$ (ratio) | T$_{Baseline}$ | 0.38 ± 0.05 | 0.37 ± 0.03 | | -0.04 to 0.03 | 0.67 |
| | T$_{2h}$ | 0.4 ± 0.07 | | 0.31 ± 0.06$^†$ | -0.15 to -0.04 | <0.001 |
| Cardiac index (l min$^{-1}$ kg$^{-1}$) | T$_{Baseline}$ | 2.3 ± 0.2 | 2.4 ± 0.5 | | -0.2 to 0.4 | 0.5 |
| | T$_{2h}$ | 2.4 ± 0.5 | | 2.3 ± 0.4 | -0.54 to 0.25 | 0.46 |
| Stroke volume variation (%) | T$_{Baseline}$ | 7 ± 3.2 | 8.3 ± 2.3 | | -0.95 to 3.5 | 0.25 |
| | T$_{2h}$ | 8.3 ± 2.9 | | 11.6 ± 2.8$^†$ | 0.92 to 5.5 | 0.008 |

Values are presented as the mean ± SD.

[a]P-values are from Student's t-test.

$^†$P<0.05 compared with T$_{Baseline}$ (paired t-test). VCV, volume-controlled ventilation; EF-initiated PC-IRV, expiratory flow-initiated pressure-controlled inverse ratio ventilation; T$_{Baseline}$, 20 min after the initial setting; T$_{2h}$, 2 h after randomisation; PaO$_2$/FiO$_2$, partial pressure of oxygen in arterial blood/fraction of inspiratory oxygen; VD$_{phys}$/V$_{TE}$, physiological dead space/expired tidal volume.

**Table 4. Changes in serum cytokine levels.**

| | Time period | VCV group | EF-initiated PC-IRV group | [a]P-value |
|---|---|---|---|---|
| IL-6 (pg ml$^{-1}$) | $T_{Baseline}$ | 8 [8–8] | 8 [8–8] | - |
| | $T_{End}$ | 31 [24.5–59] | 34 [20.5–63.5] | 0.84 |
| IL-8 (pg ml$^{-1}$) | $T_{Baseline}$ | 8 [8–8] | 8 [8–8] | - |
| | $T_{End}$ | 8 [8–9] | 8 [8–9.75] | 0.62 |

Values are presented as the median [first and third quartiles].

[a]P-values are from The Mann–Whitney U test. VCV, volume-controlled ventilation; EF-initiated PC-IRV, expiratory flow-initiated pressure-controlled inverse ratio ventilation; $T_{Baseline}$, 20 min after the initial setting; $T_{End}$, end of surgery.

(r = 0.85, P<0.001) (Fig 2), and the duration of the Trendelenburg position and pneumoperitoneum in both the EF-initiated PC-IRV group (r = 0.83, P<0.001) and the VCV group (r = 0.88, P<0.001).

The $VD_{phys}/V_{TE}$ ratio in the EF-initiated PC-IRV group was significantly smaller than that in the VCV group at $T_{2h}$ (Table 2). The $VD_{phys}/V_{TE}$ ratio at $T_{2h}$ was also negatively correlated with preoperative FVC (% predicted) (r = -0.85; 95% CI -0.95 to -0.56; $P$<0.001) and $C_{stat}$ (r = -0.63; 95% CI -0.88 to -0.12; $P$ = 0.02) in the VCV group; however, the $VD_{phys}/V_{TE}$ ratio was not correlated with either parameter in the EF-initiated PC-IRV group FVC (% predicted) (r = 0.15; 95% CI -0.42 to 0.6; $P$ = 0.62) and $C_{stat}$ (r = 0.17; 95% CI -0.39 to 0.65; $P$ = 0.55) (Fig 3).

Two patients became difficult to ventilate and required permissive hypercapnia in the VCV group, $PaCO_2$ over 50 mmHg with plateau pressure of 30 $cmH_2O$, respiratory rate of 18 bpm. One patient with a low FVC (% predicted) of 79% required permissive hypercapnia ($PaCO_2$ = 60.7 mmHg) during the entire Trendelenburg position and $CO_2$ pneumoperitoneum period. This patient had a $VD_{phys}/V_{TE}$ of 59%, $C_{stat}$ of 13 ml cm $H_2O^{-1}$, driving pressure of 24 cm $H_2O^{-1}$, respiratory rate of 18 bpm, and $V_{TE}/IBW$ of 4.9 ml kg$^{-1}$ at $T_{2h}$. Another VCV patient

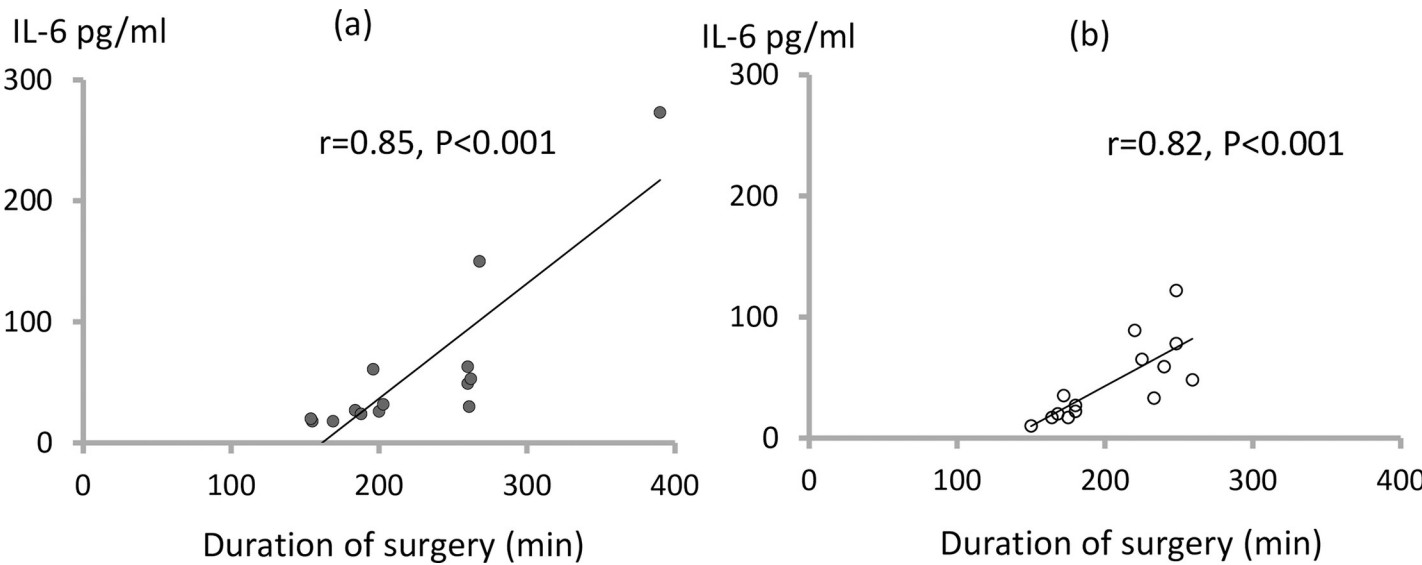

**Fig 2.** Correlation between serum IL-6 at $T_{End}$ and the duration of surgery in the volume-controlled ventilation (a) and expiratory flow-initiated pressure-controlled inverse ratio ventilation groups (b).

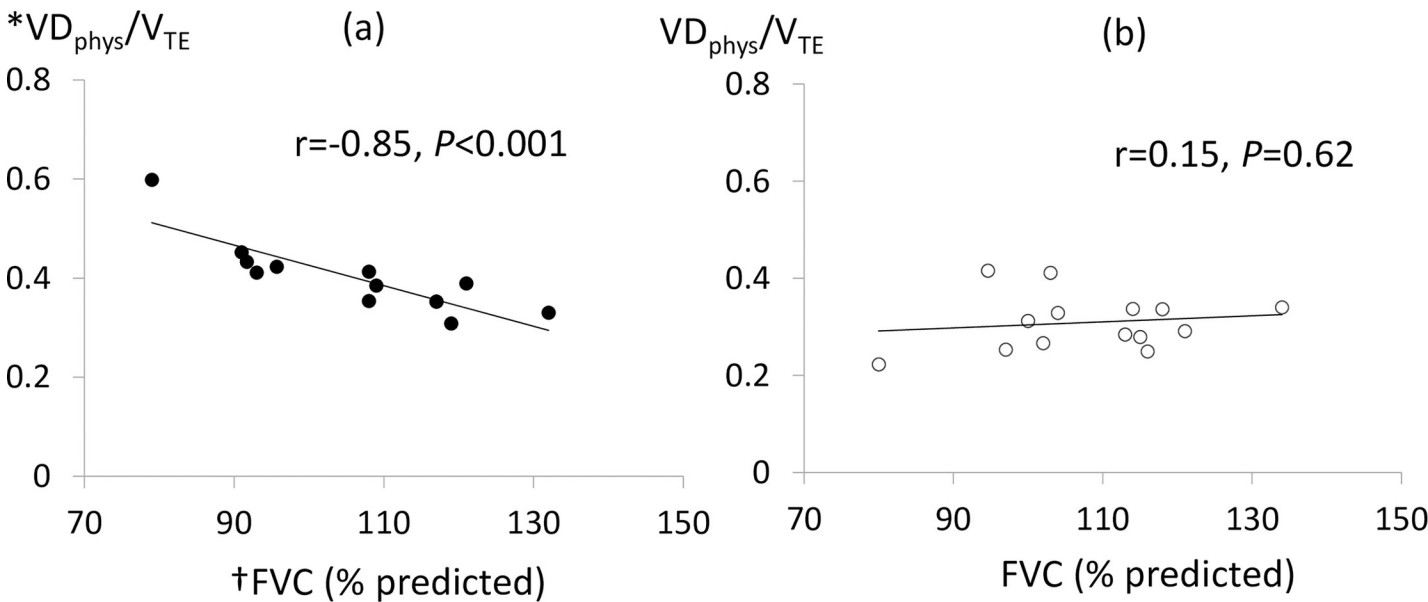

**Fig 3.** Correlation between forced vital capacity (% predicted) and physiological dead space/expired tidal volume at $T_{2h}$ in the volume-controlled ventilation (a) and expiratory flow-initiated pressure-controlled inverse ratio ventilation groups (b). $^*VD_{phys}/V_{TE}$, physiological dead space/expired tidal volume; $^†$FVC (% predicted), forced vital capacity (% predicted).

with an FVC (% predicted) of 92% had a $VD_{phys}/V_{TE}$ of 45%, $C_{stat}$ of 22 ml cm $H_2O^{-1}$, driving pressure of 20 cm $H_2O^{-1}$, respiratory rate of 14 bpm, $V_{TE}$/IBW of 6.4 ml $kg^{-1}$, and $PaCO_2$ of 49.3 mmHg at $T_{2h}$. However, this patient required permissive hypercapnia ($PaCO_2$ = 52.7 mmHg) four hours after the Trendelenburg position and $CO_2$ pneumoperitoneum period. Permissive hypercapnia was not observed in any of the patients in the EF-initiated PC-IRV group. Despite the low FVC (% predicted) value of 80% and low $C_{stat}$ of 16 ml cm $H_2O^{-1}$, it was easy to ventilate with a $VD_{phys}/V_{TE}$ of 22%, $V_{TE}$/IBW of 6.2 ml $kg^{-1}$, and $PaCO_2$ of 41 mmHg.

No instances of respiratory complications were recorded.

## Discussion

There were no significant differences in serum cytokines change between the two ventilation strategies. The expired tidal volume fell to about $\leq 6$ ml $kg^{-1}$ (IBW) at a plateau pressure of 30 cmH_2O in patients with a low FVC (% predicted), and this led to permissive hypercapnia, which may have been lung-protective. Both VCV and EF-initiated PC-IRV strategies were conducted with a plateau pressure upper limit of 30 cmH_2O, and there were no significant differences in expired tidal volume, plateau pressure, and driving pressure between the groups. Thus, the mechanical stress to alveoli is comparable, leading to an equivalent lung-protective property. Although PC-IRV prolongs the plateau time, it does not seem to affect mechanical stress when the tidal volume, plateau pressure, and driving pressure are the same condition. Moreover, IL-6 was correlated with the duration of surgery, and we observed small increases in IL-6, IL-8, and IL-1β in several cases when the duration of surgery was $\leq 200$ min. Michelet P et al also evaluated serum IL-6 and IL-8 to compare the lung-protective strategy with conventional ventilation in patients undergoing esophagectomy. Mean duration of surgery was 300 min, longer than our study, serum IL-6 and IL-8 increased more than our study, resulted in significant differences [14]. Serum IL-6 and IL-8 would not increase in short duration of

anaesthesia, surgery and $CO_2$ pneumoperitoneum. Further, in normal static compliance and FVC % predicted cases, there is no difficulty in the respiratory management with both VCV and EF initiated PC-IRV strategies. Therefore, evaluating serum cytokines in patients with healthy lungs who are undergoing surgery under general anaesthesia may require longer ventilation.

The EF-initiated PC-IRV strategy reduced $VD_{phys}$, but there were no other significant advantages with respect to the respiratory or haemodynamic parameters as compared with the VCV strategy. Tweed and Tan also studied the efficiency of PC-IRV for general anaesthesia during lower abdominal surgery and found minimal improvements in gas exchange [15]. In our study, most patients who underwent the VCV strategy had healthy lungs, normal peak inspiratory and plateau pressures, normal $PaCO_2$ and $PaO_2$, and no need to increase the respiratory rate despite the increased $VD_{phys}$. In the cases of normal respiratory compliance, a sufficient tidal volume compensated for increased $VD_{phys}$ and led to no differences between the VCV and PCV groups. However, difficulties in ventilation were encountered during the VCV strategy in patients with low respiratory compliance, $C_{stat}$, and FVC (% predicted) due to the increase in $VD_{phys}/V_{TE}$. These patients required permissive hypercapnia to avoid lung injury because of high peak inspiratory and plateau pressures and an increased respiratory rate. Although $C_{stat}$ and FVC (% predicted) showed a negative correlation with the $VD_{phys}/V_{TE}$ ratio during the Trendelenburg position and $CO_2$ pneumoperitoneum period in the VCV strategy, this correlation was not observed in patients who underwent the EF-initiated PC-IRV strategy. Thus, the VCV strategy considerably increased $VD_{phys}$ in patients with low respiratory compliance; therefore, EF-initiated PC-IRV may be especially effective in maintaining ventilation in patients with low respiratory compliance during robot-assisted laparoscopic radical prostatectomy.

PCV has been used to reduce lung injury associated with increases in peak inspiratory pressure by VCV in patients with acute respiratory distress syndrome and low respiratory compliance [16]. However, the peak inspiratory pressure was lower in the PCV group than that in the VCV group in patients with healthy lungs who were undergoing laparoscopic surgery and robot-assisted laparoscopic radical prostatectomy under general anaesthesia. Still, there were no advantages in maintaining respiratory mechanics or haemodynamics [17–20]. Moreover, Cadi and colleagues reported that lung oxygenation was improved in patients who underwent PCV as compared with that in patients who underwent VCV owing to better alveolar recruitment in patients with obesity during laparoscopic surgery; however, PCV did not reduce $VD_{phys}$ [21]. A respiratory rate of 18 beats $min^{-1}$ and an I:E ratio of 1:2 were employed, and this led to a short plateau duration and potentially lower improvement of dead space with PCV. Our findings indicate that PCV is more effective than VCV in patients with low respiratory compliance; however, PCV requires a sufficient plateau time to reduce the dead space. A prolonged inspiratory plateau time enhances gas diffusion from the alveoli into the airway, and this ultimately improves extra alveolar-derived pulmonary ventilation/perfusion (V/Q) mismatch $= \infty$ [13, 22, 23]. It also sufficiently expands the slow opening alveoli and enhances gas diffusion from the pulmonary artery into alveoli, which leads to improvements in inspiratory time-dependent V/Q mismatch $<1$ [13]. The short expiratory time in conventional PC-IRV may present a risk of lung hyperinflation and circulatory depression [9]. EF-initiated PC-IRV maintains a moderate total PEEP level with a lower risk of dynamic pulmonary hyperinflation. Moderate total PEEP prevents atelectasis and improves V/Q mismatch $= 0$ [13]. Therefore, EF-initiated PC-IRV provides an adequate expiratory time, a moderately sufficient inspiratory time, improves V/Q mismatch, effectively reduces $VD_{phys}$, and decreases the risk of dynamic pulmonary hyperinflation.

Our study had several limitations. First, both VCV and EF-initiated PC-IRV strategies were conducted with a plateau pressure upper limit of 30 cmH$_2$O, which led to a comparable alveolar response to mechanical stress. However, EF-initiated PC-IRV reduced VD$_{phys}$, which may lead to a reduction in tidal volume, plateau pressure, and driving pressure. Second, this study included patients with normal respiratory compliance for a short duration of surgery; therefore, there were small differences between the VCV and EF initiated PC-IRV strategies. Further studies that include patients with low respiratory compliance under long-term general anaesthesia and ventilatory strategies that are targeted with $PaCO_2$ and $PaO_2$ but not tidal volume nor plateau pressure are warranted to determine the utility and lung-protective properties of EF-initiated PC-IRV.

## Conclusions

There were no differences in the lung-protective properties between the VCV and EF-initiated PC-IRV strategies with a plateau pressure upper limit of 30 cmH$_2$O. However, VCV increased VD$_{phys}$ and sometimes required permissive hypercapnia to prevent high plateau pressure in patients with a low FVC (% predicted). EF-initiated PC-IRV reduced VD$_{phys}$ and facilitated effective $CO_2$ elimination; thus, it may be useful for reducing the ventilatory volume that is necessary to maintain normocapnia in low respiratory compliance situations, such as the Trendelenburg position and $CO_2$ pneumoperitoneum, in patients with a low FVC (% predicted).

## Supporting information

**S1 Checklist. CONSORT checklist.**
(DOCX)

**S1 File. Japanese protocol submitted to IRB.**
(DOCX)

**S2 File. English protocol.**
(DOCX)

**S1 Data. Dataset.**
(XLSX)

## Acknowledgments

The authors would like to thank Editage (www.editage.com) for English language editing.

## Author Contributions

**Conceptualization:** Go Hirabayashi, Tomio Andoh.

**Data curation:** Go Hirabayashi, Minami Saito, Sachiko Terayama, Yuki Akihisa, Koichi Maruyama.

**Formal analysis:** Go Hirabayashi.

**Investigation:** Go Hirabayashi.

**Methodology:** Go Hirabayashi, Tomio Andoh.

**Project administration:** Go Hirabayashi.

**Supervision:** Tomio Andoh.

**Writing – original draft:** Go Hirabayashi.

**Writing – review & editing:** Tomio Andoh.

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
