## [Decision Letter · Decision Letter 0]

1 Aug 2020

PONE-D-20-10447

Lung-protective properties of expiratory flow-initiatedpressure-controlled inverse ratio ventilation:  A randomised controlled trial

PLOS ONE

Dear Dr. Hirabayashi,

Thank you for submitting your manuscript to PLOS ONE. After careful consideration, we feel that it has merit but does not fully meet PLOS ONE’s publication criteria as it currently stands. Therefore, we invite you to submit a revised version of the manuscript that addresses the points raised during the review process.

Specifically, the reviewers have raised overlapping concerns about the reporting of the statistical methodology and study design in the manuscript.

We look forward to receiving your revised manuscript.

Kind regards,

Richard Hodge

Associate Editor

PLOS ONE

Journal Requirements:

2.During our internal evaluation, the in-house editorial staff noted that the clinical trial was registered at ClinicalTrials.gov

'It was approved by the Ethical Committee of Teikyo University School of Medicine, Tokyo, Japan on 12 September 2017 (No. 17-063) and was registered at ClinicalTrials.gov UMIN000029552' (line 82-84).

However, the trial registration provided in the manuscript is a UMIN Clinical Trials Registry number. At this time, we ask that you also provide the ClinicalTrias.gov trial registration number in your Methods section.

3. In the Methods section, please confirm that the study protocol provided in Supplementary Figure 1 is the original study protocol that was specifically submitted to and approved by the Ethical Committee of Teikyo University School of Medicine.

4. In your Methods section, please provide additional information about the participant recruitment method and the demographic details of your participants. Please ensure you have provided sufficient details to replicate the analyses such as: a) the recruitment date range (month and year).

5. Please provide a sample size and power calculation in the Methods, or discuss the reasons for not performing one before study initiation.

6.Thank you for stating the following financial disclosure:

 [The funders had no role in study design, data collection and analysis, decision to publish, or preparation of the manuscript.].

Reviewers' comments:

Reviewer's Responses to Questions

**Comments to the Author**

1. Is the manuscript technically sound, and do the data support the conclusions?

Reviewer #1: Partly

Reviewer #2: Yes

Reviewer #3: Partly

2. Has the statistical analysis been performed appropriately and rigorously? 

Reviewer #1: I Don't Know

Reviewer #2: Yes

Reviewer #3: No

3. Have the authors made all data underlying the findings in their manuscript fully available?

Reviewer #1: Yes

Reviewer #2: Yes

Reviewer #3: Yes

4. Is the manuscript presented in an intelligible fashion and written in standard English?

Reviewer #1: No

Reviewer #2: Yes

Reviewer #3: Yes

5. Review Comments to the Author

Reviewer #1: It is thought that the results of previous studies suggesting that the serum cytokines selected in this study could was used as a surrogate marker for lung injury should be included in the introduction or discussion. In addition, I think it is necessary to consider why there were no differences of cytokines between the two groups although EF-initiated PC-IRV reduce dead space. The primary endpoint of this study was not change of respiratory mechanics but the change of serum cytokines.

Page 4, 91-93. Has the same surgeon performed surgery on all enrolled patients?

Page 6, 158-160. Please add references about the cytokines selected in this study.

Page 6, 172-175. Was there data regarding serum cytokine obtained from previous studies or preliminary study for calculating the sample size? Please describe the sample size calculation in more detail.

Page 6, 175-178. What are the criteria for using non-parametric and parametric methods differently depending on the type of variable? Did you perform a normality test?

Page 9, 231-233. In this study, correlation analysis was performed between IL-6 and the duration of surgery. Did your group conduct correlation analysis between cytokine and duration of the Trendelenburg position?

Page 10, 270-272. The meaning of the sentence regarding permissive hypercapnia is not clear. Can permissive hypercapnia in two patients observed only in VCV give special significance compared to PC-IRV?

Page 11, 280-282. Isn't the duration of the Trendelenburg position and pneumoperitoneum more meaningful than simply duration of surgery?

Page 11, 303-308. The meaning of this paragraph is unclear. It was noted that the peak inspiratory pressure was lower in the PCV group than that in the VCV in patients with laparoscopy and robot surgery. Why were there advantages in maintaining respiratory mechanics?

Reviewer #2: It is well written and well designed study

These are my suggestions

It would be better to give the ventilator strategy as a scheme in to the method part

It would be better to add VDshunt/VTE , ETCO2 , PaCO2-PetCO2 in results and table 2

Is PVC statement in line 306 actually PCV ?

Reviewer #3: A prospective two-arm randomized clinical trial was conducted to compare differences in serum cytokine levels in men undergoing robot-assisted laparoscopic radical prostatectomy. No significant differences in IL-6 levels at the end of surgery were observed between the two arms.

Minor revisions:

1- Line 91: Provide specific details on how the randomization list was generated. If randomized blocks were used, indicate the block size.

2- Line 172: Provide more complete details for the statistical power calculation. The power calculation should include: sample size, alpha level (indicating one or two-sided), minimal detectable difference and statistical testing method.

3- Line 175 states, “The Mann–Whitney U test was used to compare differences in the serum

cytokine levels between the two groups.” To clarify, was the Mann-Whitney U test used to compare differences only at the end of surgery?

4- Line 187: Clarify that the patients were randomized rather than simply divided.

5- Table 1: In the statistical analysis section, state the statistical methods used to compare patient and surgery characteristics shown in Table 1. If data is normally distributed, summarize the results using mean and standard deviation. If the data is not normally distributed provide the median, first and third quartiles. Use nonparametric tests (Mann-Whitney U-test) to compare groups when the distribution of the data is not normally distributed. Use parametric tests (t-tests) to compare groups with normally distributed data.

6- Table 2: Tests of the interaction of arm by time is more appropriate than repeatedly applying t-tests when comparing the data shown in Table 2.

7- Line 232: Clarify the rs notation. There appears to be a typographical error “rs=0,082”.

8- The p-value associated with a correlation is a test of the null hypothesis: correlation equal to zero; however, the absolute magnitude of the coefficient indicates the strength of the linear relationship between two variables. In general, the strength or correlation coefficient is the more important statistic to reflect upon.

Below is a table for interpreting correlation coefficients:

Coefficient (absolute value) Interpretation

0.90 - 1.0 Very Strong

0.70 - 0.89 Strong

0.40 - 0.69 Moderate

0.10 - 0.39 Weak

less than 0.10 Negligible correlation

9- Indicate if adverse events were collected according to a standardized method.

10- Add the correlation coefficients to Figures 2 and 3.

11- Indicate the funding source(s), and the role of the funder(s)?

12- All acronyms and abbreviations must be spelled out in first use.

6. PLOS authors have the option to publish the peer review history of their article (what does this mean?). If published, this will include your full peer review and any attached files.

Reviewer #1: No

Reviewer #2: No

Reviewer #3: No

---

## [Author Response · Author response to Decision Letter 0]

26 Aug 2020

Thank you very much for your letter dated August 1. On behalf of all of the authors, I would like to re-submit our revised manuscript titled “Lung-protective properties of expiratory flow-initiated pressure-controlled inverse ratio ventilation: A randomised controlled trial” (manuscript ID: PONE-D-20-10447) for publication in PLOS One.

We would like to thank you and the reviewers for the helpful comments, which we feel have helped us improve our manuscript.

---

## [Decision Letter · Decision Letter 1]

19 Oct 2020

PONE-D-20-10447R1

Lung-protective properties of expiratory flow-initiated pressure-controlled inverse ratio ventilation:  A randomised controlled trial

PLOS ONE

Dear Dr. Hirabayashi,

Thank you for submitting your manuscript to PLOS ONE. After careful consideration, we feel that it has merit but does not fully meet PLOS ONE’s publication criteria as it currently stands. Therefore, we invite you to submit a revised version of the manuscript that addresses the points raised during the review process.

We look forward to receiving your revised manuscript.

Kind regards,

Steven Eric Wolf, MD

Academic Editor

PLOS ONE

Additional Editor Comments (if provided):

Editor - Thank you for resubmitting your paper. As promised, I sent it back to the original referees who are now almost completely satisfied save a few minor issues. Please carefully consider the comments below and reply directly to each in a cover letter with appropriate marked and linked changes to the manuscript. I look forward to receiving the next version which I will handle personally for timeliness.

If data is normally distributed, summarize using mean (SD) and compare using parametric methods, possibly t-tests. However, if data is non-parametic summarize using median (first and third quartiles) and compare using non-parametic method such as Mann Whitney U tests.

Reviewers' comments:

Reviewer's Responses to Questions

**Comments to the Author**

1. If the authors have adequately addressed your comments raised in a previous round of review and you feel that this manuscript is now acceptable for publication, you may indicate that here to bypass the “Comments to the Author” section, enter your conflict of interest statement in the “Confidential to Editor” section, and submit your "Accept" recommendation.

Reviewer #1: (No Response)

Reviewer #2: All comments have been addressed

Reviewer #3: (No Response)

2. Is the manuscript technically sound, and do the data support the conclusions?

Reviewer #1: (No Response)

Reviewer #2: Yes

Reviewer #3: Yes

3. Has the statistical analysis been performed appropriately and rigorously? 

Reviewer #1: (No Response)

Reviewer #2: N/A

Reviewer #3: Yes

4. Have the authors made all data underlying the findings in their manuscript fully available?

Reviewer #1: (No Response)

Reviewer #2: Yes

Reviewer #3: Yes

5. Is the manuscript presented in an intelligible fashion and written in standard English?

Reviewer #1: (No Response)

Reviewer #2: Yes

Reviewer #3: Yes

6. Review Comments to the Author

Reviewer #1: (No Response)

Reviewer #2: (No Response)

Reviewer #3: If data is normally distributed, summarize using mean (SD) and compare using parametric methods, possibly t-tests. However, if data is nonparametic summarize using median (first and third quartiles) and compare using nonparametic method such as Mann Whitney U tests.

7. PLOS authors have the option to publish the peer review history of their article (what does this mean?). If published, this will include your full peer review and any attached files.

Reviewer #1: No

Reviewer #2: No

Reviewer #3: No

---

## [Author Response · Author response to Decision Letter 1]

31 Oct 2020

We thank you and the reviewers for your thoughtful suggestions and insights. The manuscript has benefited from this valuable feedback. We have reviewed our manuscript and made necessary changes in accordance with the reviewers’ suggestions. We have also prepared point-by-point responses to all comments and have attached them herewith. All revisions are marked using Tracked Changes in the revised manuscript and although we have also not listed all minor changes made to the manuscript in the rebuttal letter, none of those changes has modified the content, conclusions, or framework of the paper.

We look forward to working with you and the reviewers to move this manuscript closer to publication in PLOS One.

---

## [Editor Report · Decision Letter 2]

2 Dec 2020

Lung-protective properties of expiratory flow-initiated pressure-controlled inverse ratio ventilation:  A randomised controlled trial

PONE-D-20-10447R2

Dear Dr. Hirabayashi,

We’re pleased to inform you that your manuscript has been judged scientifically suitable for publication and will be formally accepted for publication once it meets all outstanding technical requirements.

Kind regards,

Steven Eric Wolf, MD

Academic Editor

PLOS ONE
---

## [Editor Report · Acceptance letter]

9 Dec 2020

PONE-D-20-10447R2 

Lung-protective properties of expiratory flow-initiated pressure-controlled inverse ratio ventilation: A randomised controlled trial 

Dear Dr. Hirabayashi:

I'm pleased to inform you that your manuscript has been deemed suitable for publication in PLOS ONE. Congratulations! Your manuscript is now with our production department. 

Kind regards, 

on behalf of

Dr. Steven Eric Wolf 

Academic Editor

PLOS ONE